# Intra-Articular Mesenchymal Stem Cell Injection for Knee Osteoarthritis: Mechanisms and Clinical Evidence

**DOI:** 10.3390/ijms24010059

**Published:** 2022-12-21

**Authors:** Pengxu Wei, Ruixue Bao

**Affiliations:** 1Beijing Key Laboratory of Rehabilitation Technical Aids for Old-Age Disability, Key Laboratory of Neuro-Functional Information and Rehabilitation Engineering of the Ministry of Civil Affairs, National Research Center for Rehabilitation Technical Aids, Beijing 100176, China; 2Key Laboratory of Biomechanics and Mechanobiology (Beihang University), Ministry of Education, Beijing Advanced Innovation Center for Biomedical Engineering, School of Biological Science and Medical Engineering, Beihang University, Beijing 100083, China; 3School of Rehabilitation Medicine, China Rehabilitation Research Center, Capital Medical University, Beijing 100068, China

**Keywords:** mesenchymal stem cells, knee osteoarthritis, intra-articular injection, dose, hyaline cartilage, cartilage repair, cartilage regeneration, human

## Abstract

Knee osteoarthritis presents higher incidences than other joints, with increased prevalence during aging. It is a progressive process and may eventually lead to disability. Mesenchymal stem cells (MSCs) are expected to repair damaged issues due to trilineage potential, trophic effects, and immunomodulatory properties of MSCs. Intra-articular MSC injection was reported to treat knee osteoarthritis in many studies. This review focuses on several issues of intra-articular MSC injection for knee osteoarthritis, including doses of MSCs applied for injection and the possibility of cartilage regeneration following MSC injection. Intra-articular MSC injection induced hyaline-like cartilage regeneration, which could be seen by arthroscopy in several studies. Additionally, anatomical, biomechanical, and biochemical changes during aging and other causes participate in the development of knee osteoarthritis. Conversely, appropriate intervention based on these anatomical, biomechanical, biochemical, and functional properties and their interactions may postpone the progress of knee OA and facilitate cartilage repair induced by MSC injection. Hence, post-injection rehabilitation programs and related mechanisms are discussed.

## 1. Introduction

Knee osteoarthritis (OA) has higher incidences than other joints, and its prevalence increases during aging, especially after age 65 [1]. Knee OA is often considered a result of “wear and tear”. Several risk factors worsen the severity of knee OA, including knee joint injury history, being overweight and obese, repetitive loading, muscle weakness, malalignment, post-menopausal changes, and genetic predisposition [2,3].

OA is a disease of the whole joint and features inflammation of the synovial joint, loss of cartilage, bone changes of the joints, meniscus damage, and deterioration of tendons and ligaments. Radiographically, patients with OA present the formation of osteophytes, narrowing of the joint space, subchondral sclerosis, subchondral cyst formation, and chondrocalcinosis. Considerable evidence also indicates the involvement of the infrapatellar fat pad in the development of knee OA [4,5,6].

Knee OA is a progressive process and may eventually lead to disability. Symptoms of the disease include pain, stiffness, swelling, decreased range of movement of the knee joint, and joint deformity. Therapies for knee OA include physiotherapy, nonsteroidal anti-inflammatory drugs, pain-relieving agents, hyaluronic acid, and intra-articular corticosteroid injections. All these treatments may relieve symptoms and slow the progression of the condition but cannot repair damaged articular cartilage. Consequently, total knee arthroplasty may eventually be needed [7,8].

Cartilage defects have a limited intrinsic healing capacity. Small defects can spontaneously undergo repair with the production of hyaline cartilage. However, large defects undergo repair only with the production of fibrous tissue or fibrocartilage, which is biochemically and biomechanically different from normal hyaline cartilage. Therefore, degeneration occurs subsequently and can progress to osteoarthritic changes in some cases [9]. Mesenchymal stem cells (MSCs) have self-renew and multidirectional-differentiation potentials and can regulate immunity, resist inflammation, facilitate angiogenesis, and improve regeneration [7,8]. MSCs can be isolated from multiple tissues, such as the bone marrow, skeletal muscle tissue, synovial membranes, periodontal ligaments, Wharton’s jelly, umbilical cords, umbilical cord blood, amniotic fluid, placentae, and adipose tissue (subcutaneous, abdominal, or infrapatellar fat pad origins) [6,10]. Tissue sources of MSCs impact their effects. Synovium-derived MSCs have better chondrogenesis potential than those derived from bone marrow, periosteum, skeletal muscle, and adipose tissue [11]. MSCs derived from infrapatellar fat pad exhibit better chondrogenic potential than those derived from bone marrow or subcutaneous fat [12], and infrapatellar fat pad and synovium are preferred tissue sources of MSCs for cartilage repair in patients >60 years [13]. Intra-articular MSC injection is far less invasive than knee arthroplasty. Injected MSCs and their secretion are expected to repair damaged issues due to the trilineage potential, trophic effects, and immunomodulatory properties of MSCs [14].

## 2. Arrangement of the Review and Literature Search Strategy

This review mainly focuses on the effects of doses of MSCs for intra-articular MSC injection for knee OA. A literature search was performed in electronic databases including PubMed, EMBASE, and Cochrane Library by using the following terms: (Knee Osteoarthritis OR Osteoarthritis of Knee OR Osteoarthritis of Knees OR Knee, Osteoarthritis OR Knees, Osteoarthritis) AND (Stem Cells OR Progenitor Cells OR Mesenchymal Stromal Cells OR IPS Cells) AND (Human OR Patients). The related articles’ function was used to broaden the search results. Only studies containing different doses, substances, or treatment techniques were included. The deadline for retrieval was August 2022. We summarize studies using intra-articular MSC injection to treat knee OA in humans. Before comparing dose effects, studies comparing MSCs with other substances and works comparing MSC injection with surgery are also introduced.

This review starts with discussions of anatomical, biochemical, and biomechanical properties of knee cartilage given that these properties change during aging and other causes, which participate in the development of knee OA and are related to what programs should be applied after injection with MSCs.

## 3. Anatomical, Biochemical, and Biomechanical Properties of Knee Cartilage

Cartilaginous tissues can be divided into three types: hyaline, fibro-, and elastic cartilage. In a healthy state, hyaline cartilage allows nearly frictionless movement across its surface [15]. Cartilage is composed of chondrocytes immersed in the cartilage extracellular matrix consisting of polysaccharides, fibrous proteins, and interstitial fluid. No blood, lymphatics, and nerves directly supply hyaline cartilage, and nutrition originates on diffusion from the surrounding tissues. Hyaline cartilage (often referred to as articular cartilage) is located on the articulating surfaces of the synovial joints such as the knee and elbow. The matrix of hyaline cartilage is rich in proteoglycans (mainly aggrecans, that is, the large aggregating proteoglycan) and short and dispersed collagen (primarily type II collagen). The cartilage cells, chondrocytes, comprise 2–3% of the total cartilage volume. Fibrocartilage presents in knee menisci. Compared with hyaline cartilage, fibrocartilage is composed of a very small amount of chondrocytes and fewer proteoglycans but more type I and type II collagen. The dense collagen fibrils make fibrocartilage highly resistant to compression. Elastic cartilage within the epiglottis, auricle, and Eustachian tube contains dense elastic fibrils branched in multiple directions and is elastic/flexible to resist repeated bending [16,17].

Adult hyaline cartilage is often introduced to contain four distinct layers/zones, namely, the superficial/gliding, middle, deep, and calcified zones (Figure 1). These zones can be distinguished via the shape and orientation of the chondrocytes and the distribution of type II collagen [15]. From the surface to the deeper zone in the matrix, the fibrillar and interstitial fluid component decrease whereas proteoglycans increase [16]. The calcified zone is the product of endochondral ossification and persists after growth plate closure. A tidemark, as a histologically defined boundary, is located between hyaline cartilage and the thin calcified cartilage. Nutrition is provided by diffusion from the synovial fluid above the tidemark and by the underlying vascular supply from the subchondral bone below the tidemark. The subchondral bone lies beneath the calcified cartilage. The subchondral bone marrow is below the subchondral bone [15].

In healthy young adults, the thickness of different regions of hyaline cartilage (tibiofemoral joint) scanned by magnetic resonance imaging (MRI) ranges from 1.4 ± 0.4 mm to 4.0 ± 0.9 mm [18]. The cartilage thickness is thicker in the femur than in the corresponding tibia regions, and the patella cartilage is thicker than the tibiofemoral joint cartilage [19]. MRI-measured cartilage thicknesses are up to 4.5 ± 1.3 mm for the tibiofemoral joint and 5.2 ± 1.3 mm for the patella in people aged 7–18 years old [20].

Loading and aging influence cartilage thickness. The non-load-bearing areas of the femur cartilage are thicker than the load-bearing areas, and cartilage thinning occurs in all regions (the femur, tibia, and patella) during aging [19].

The articular cartilage has an outermost layer called the lamina splendens (Figure 1), which corresponds to the very thin surface amorphous layer observed using a cryoscanning electron microscope and is thus not an “optical effect”. The lamina splendens has a thickness on the scale of 10 μm (from 80 μm in an 11-year-old boy to 13–28 μm in a 62-year-old woman) [21]. The lamina splendens is a layer resting at the very surface of the superficial/gliding zone and (together with lubricin [22,23]) provides a nearly frictionless surface during sliding between hyaline cartilages; it has a friction coefficient at the level of 10^−3^ under various physiological pressures (much less than the friction coefficient between polished stainless steel and ice, which is 0.02–0.06) [15,24,25]. Lubricin is a highly glycosylated mucin-like glycoprotein and plays a critical role in lubricating inter-cartilage boundaries [23,26,27].

The extremely low friction of the hyaline cartilage surface relies on the normal status of the joint surface (e.g., morphology) and synovial fluid [28]. Aging and/or lesions cause damage to the outer surface of articular/hyaline cartilage and subsequently lead to increasing inter-cartilage friction, which initiates knee OA [25].

The superficial zone, middle zone, and deep and calcified zone form approximately 10–20%, 40–60%, and 30–40% of the full thickness of adult cartilage, respectively. The type II collagen fibrils in the deep zone are approximately twice as thick as those in the superficial zone (40–80 nm vs. 30–35 nm) [25].

Notably, collagen fibrils within the superficial zone often orient in a direction parallel to the surface of the articular surface [29] and the lamina splendens do not contain collagen fibrils [21]. However, the 3D imaging technique reveals interwoven collagen bundles within the lamina splendens and obliquely oriented collagen fibrils within the superficial zone. These parallel-, obliquely, and perpendicularly oriented collagen fibrils constitute a 3D collagen network/scaffold, which anchors the articular (hyaline) cartilage to the subchondral bone (Figure 1). Collagen has great tensile strength; the 3D network can also constrain the swelling pressure of hydrated proteoglycans in the cartilage matrix and form a buffer/cushion. The interwoven collagen bundles running within the lamina splendens provide tensile strength to resist wear and shear stresses. These protective mechanisms are damaged during aging or after disruption of the articular surface due to gradually disappearing interwoven collagen bundles and a changing alignment of collagen fibrils (from an oblique orientation to a direction perpendicular to the articular surface) [29].

Proteoglycans make up about 25% of the dry weight of hyaline cartilage. Proteoglycan in cartilage matrix is mainly aggrecan, which tends to aggregate into large supramolecular complexes >200,000,000 Dalton together with hyaluronan and link protein (by contrast, hen ovalbumin is a 43,000 Dalton glycoprotein [30]). The core protein of an aggrecan contains an extended region attaching >100 chondroitin sulfate chains, which are negatively charged glycosaminoglycan (GAG) chains. The negatively charged forces provide aggrecan with the ability to attract and retain water molecules. Electrostatic repulsion forces generated by the numerous negatively charged chondroitin sulfate chains also repel each other and thus extend/inflate the structure of aggrecan, which provides cartilage an anti-compressive feature. Such an osmotic swelling pressure in human femoral heads is estimated at a magnitude of approximately 0.2 MPa (i.e., 5 kg-force per square centimeter) [31,32].

Collagen (90–95% are type II) contributes to approximately 60% of the dry weight of articular cartilage [25], and it can only be elongated less than 10% of its total length; much of the increase originates from the straightening of the fibers rather than true elongation [33]. The swelling pressure to inflate the matrix is resisted by strong tension generated by the collagen fibril network [31], given the fact that removing collagen fibrils converts the gel-like vitreous into a viscous liquid [34,35]. The balanced forces allow a gel-like feature of the cartilage matrix [29,36,37]. Notably, the electrostatic repulsion of negative charges rather than water contributes to compressive resistance [36].

Glycosaminoglycans (GAGs), as the main component of proteoglycans, reach the highest concentration in the middle layer and the lowest concentration at the surface of hyaline cartilage. As a result, the superficial layer has the lowest osmotic pressure generated by hydrated proteoglycans, which is resisted by the tension of collagen fibrils. Otherwise, the collagen fibrils have to be under exceptionally high tension to oppose a very high osmotic pressure gradient. This orientation is in line with that of collagen fibrils in different layers: parallel-orientated collagen fibrils at the surface resist shear forces; oblique and perpendicular collagen fibrils in the deeper layers can stand expansion forces from hydrated proteoglycans. Changes in the concentration of GAGs and damaged collagen fibril network due to aging/osteoarthritic diseases can result in the vulnerability of hyaline cartilage [36,37].

Most intra-articular MSC injection studies focused on hyaline cartilage. However, the meniscus is another important shock absorber for knee joints in that the thickest parts of the meniscus are greater than the sum of the thickness of hyaline cartilage of the femoral condyle and tibial plateau. Force absorption during joint loading is achieved by meniscus and hyaline cartilage [38]. Aging or joint injuries often lead to damage in the meniscus and hyaline cartilage. A meniscus resection of only 10% of the entire volume can contribute to the development of chondral lesions [39], and the incidence of OA increases up to sevenfold for patients who receive a meniscectomy [40]. An increased meniscal volume following intra-articular MSC injection could be identified by MRI [41]. Thus, the knee meniscus needs to be given attention. In a study, intra-articular MSC injection was combined with a torn-meniscus injection in addition to other damaged tissues [42].

The knee meniscus presents a wedge formation between femoral condyles and the tibial plateau to increase knee joint stability and absorb shock forces during joint loading. The thickness of the medial meniscus is 5.2–6.9 mm. The thickness of the anterior third of the lateral meniscus is 3.8–4.73 mm, and the thickness of its middle third/posterior third is 5.9–6.5 mm/5.3–6.2 mm [43]. The knee meniscus contains 65–72% of water, 20–25% of collagen, and <1% proteoglycans. The extracellular matrix consists of proteoglycans and a densely interwoven collagen network (mainly type I collagen). Proteoglycans (mainly aggrecan) absorb water, which provides the viscoelastic anti-compression property of the meniscus. The meniscus surface, which consists of randomly oriented collagen fibrils, is smooth to minimize friction. By contrast, the deep layer contains circumferentially oriented collagen fibrils to convert compressive force into circumferential stresses during joint loading. A few radially oriented collagen fibrils, together with the circumferential fibrils, cross the deep layer to constitute an interwoven network for resisting splitting [43,44,45,46].

In summary, the biochemical properties of knee cartilage are closely related to the anatomical and biomechanical features of these structures. Anatomical, biochemical, and biomechanical changes during aging and other causes are involved in the development of knee OA. Meanwhile, appropriate intervention based on these properties may postpone the progress of knee OA and facilitate cartilage repair induced by MSC injection.

## 4. Studies Comparing MSCs with Other Substances

Several studies compared the effects of intra-articular injection with MSCs and other substances (Table 1, Table 2 and Table 3).

Three studies compared the effects of saline and MSCs; one found similar relief of pain in bone marrow aspirate concentrate (containing MSCs and hematopoietic stem cells) and saline injection groups but injected cells were mainly hematopoietic stem cells (34,400 MSCs vs. 4,620,000 hematopoietic stem cells) [47]. The two other studies, which adopted autologous adipose-derived MSCs or allogenic placenta-derived MSCs, found that the MSCs injection group had better clinical symptom improvements and less cartilage defect/better chondral thickness improvement than the saline injection group [48,49]. Overall, MSC injection in a dose of 0.5–0.6 × 10^8^ and 1.0 × 10^8^ cells demonstrated better effects than saline injection [48,49], whereas a low dose (34,400 MSCs) did not [47].

Hyaluronic acid is often applied to treat knee OA. Several studies compared the effects of hyaluronic acid and MSCs. Better functional improvement and MRI-identified cartilage quality improvements were found in the MSCs group when compared with MSCs or hyaluronic acid alone [50]. A study compared three groups, namely, intra-articular knee injections of hyaluronic acid at baseline and 6 months, umbilical cord-derived MSCs at baseline and 6 months, and umbilical cord-derived MSCs alone at baseline. The results showed that the twice MSC injection group reached significantly lower levels of pain and better knee function than the hyaluronic acid group [51]. Other studies also found a higher MRI-identified increase in knee articular cartilage in the MSCs group than in the hyaluronic acid group, in addition to significantly greater improvements in clinical symptoms in the MSCs group than in the HA group [52,53]. Thus, MSC injection alone might better relieve knee OA symptoms and lead to better cartilage repair than the hyaluronic acid injection alone.

Platelet-rich plasma (PRP) is increasingly used for injection treatment in patients with knee OA. Many anabolic growth factors (e.g., FGF, TGF-β1, TGF-β2, and EGF) and anti-inflammatory cytokines (e.g., IL-1ra, sTNF-R1, sTNFRII, IL-4, IL-10, IL-13, and IFNγ) can be found in PRP. These substances play different roles in modifying the pathological process of knee OA [54,55,56,57]. No studies are currently available to compare the effects of PRP alone with those of MSCs alone but several studies compared the effects of MSCs + PRP with PRP alone in patients with knee OA. The MSCs + PRP group had better pain reduction and functional improvements than MSC injection alone [58]. In another study, only the MSCs + PRP group resulted in pain reduction and functional improvements [54].

Corticosteroid injection is often applied to treat knee OA patients. A study found that MSCs alone and MSCs + PRP groups had better therapeutic effects (improvements assessed with the Knee Injury and Osteoarthritis Outcome Score) than the corticosteroid group [59].

On the basis of such findings of the controlled studies, intra-articular injection of MSCs demonstrates better effects than hyaluronic acid alone and corticosteroid injection alone. MSCs + PRP also can lead to a higher level of pain reduction and functional improvements in patients with knee OA. However, definite conclusions cannot be drawn due to only a few studies available and the heterogeneity of these works.

**Table 1 ijms-24-00059-t001:** Substance comparison injection studies using allogeneic MSCs.

Cell Type	Cell Dosage	Cell Passage	Combined Interventions	Control with Non-MSC Agents	Knee OA Grading	Injection Time	Follow Up	Outcome	Reference
Allogenic PDMSCs	0.5–0.6 × 10^8^	12	No	Normal saline		1	24 weeks	Range of motion improvement and pain reduction until 8 weeks. Chondral thickness improved at 24 weeks, and anterior cruciate ligament healing may be observed, but no meniscus repair was detected by MR arthrography.	[49]
Allogeneic BMMSCs	40 × 10^6^	3	No	Hyaluronic acid alone	KL II–IV	1	12 months	Better functional improvement and cartilage quality improvements by MRI in the MSCs group.	[50]
Allogenic UCMSCs	20 × 10^6^	5	Avoid physical activity for 48 h after the procedure.	Hyaluronic acid (0 + 6 months)	KL II–III	1 or 2 (0 + 6 months)	12 months	Pain reduction and function improvement were only observed in the repeated MSC injection group.	[51]

BMMSCs, bone marrow-derived MSCs; PDMSCs, placenta-derived MSCs; UCMSCs, umbilical cord-derived MSCs; KL, the Kellgren and Lawrence OA classification.

**Table 2 ijms-24-00059-t002:** Substance comparison injection studies using autologous bone marrow-derived MSCs.

Cell Type	Cell Dosage	Cell Passage	Combined Interventions	Control with Non-MSC Agents	Knee OA Grading	Injection Time	Follow Up	Outcome	Reference
Autologous BMAC	34,400 MSCs + 4,620,000 HSCs	0	Platelet-poor plasma (to increase injection volume). No brace and physical therapy provided.	Saline injection into the other knee with OA	KL < IV	1	6 months	Similar relief of pain in BMAC- and saline-treated arthritic knees.	[47]
Autologous BMMSCs	100 × 10^6^	Unavailable	PRP (3 times)	PRP (3 times) alone	KL II–IV	1	12 months	Only the MSCs + PRP had pain reduction and functional improvement.	[54]
Autologous BMMSCs	40 × 10^6^	≤2	Drugs, hydrotherapy, heat, and ultrasound or acupuncture were prohibited.	MSCs + PRP	KL II–IV	1	12 months	Both groups had improvements, but MSCs + PRP induced better effects.	[58]
Autologous BMMSCs	40 × 10^6^	≤2	PRP	Corticosteroid	KL I–IV	1	12 months	MSCs and MSCs + PRP groups showed the highest percentage of improvement compared with the corticosteroid group.	[59]
Autologous BMMSCs	2740–7540 × 20	0	Instructions for immediate full weight-bearing. Physical therapy was considered unnecessary.	Implantation in the subchondral bone of the medial femur and tibia	KL I–IV	1	15 years	Both groups resulted in pain relief, but time conversion to total knee arthroplasty was longer in those receiving subchondral MSC injections.	[60]

BMAC, bone marrow aspirate concentrate, containing MSCs and hematopoietic stem cells (HSCs); BMMSCs, bone marrow-derived MSCs; KL, the Kellgren and Lawrence OA classification; PRP, platelet-rich plasma.

**Table 3 ijms-24-00059-t003:** Substance comparison injection studies using autologous adipose-derived MSCs.

Cell Type	Cell Dosage	Cell Passage	Combined Interventions	Control with Non-MSC Agents	Knee OA Grading	Injection Time	Follow Up	Outcome	Reference
Autologous ADMSCs	1 × 10^8^	Unavailable	No	Normal saline	KL II–IV	1	6 months	Pain reduction and functional improvement only observed in the MSCs group. Worse cartilage defect by MRI only in the control group.	[48]
Autologous ADMSCs	5 × 10^7^ (0 + 3rd week)	Unavailable	Rest for 24 h following each injection.	Hyaluronic acid (1/week for 4 weeks)	KL I–IV			Higher increase in articular cartilage volume by MRI in the MSCs group.	[52]
Autologous ADMSCs	8 × 10^6^	Unavailable	Avoid weight-bearing motions on the affected knee, such as standing for prolonged periods, jogging, and lifting heavy objects during the first 3 days.	Hyaluronic acid	KL I–IV	1	12 months	Greater improvements observed in the MSCs group.	[53]
Autologous ADMSCs	100 × 10^6^ (single injection)/ 100 × 10^6^ (baseline + 6 month)	2	None for the control group. The MSCs group remained non-weight-bearing and used crutches for 4 weeks. A range of motion and quadriceps exercises were also provided.	Conventional conservative management only	KL II–III	1 or 2	12 months	Better functional improvement and pain reduction were observed in the MSCs group.	[61]

ADMSCs, adipose-derived MSCs; KL, the Kellgren and Lawrence OA classification.

## 5. Can MSC Injection Induce Cartilage Regeneration?

Currently, MSC injection for treating knee OA is considered to have several mechanisms, including immune or inflammatory modulation effects and tissue repair or regeneration. Whether MSCs hold a capacity in an in vivo condition to repair and regenerate cartilaginous tissue in the joint is unclear [62,63].

Whether intra-articular injection of MSCs can induce cartilage regeneration is interesting and crucial. When compared with autologous chondrocyte implantation, autologous synovia-derived MSC implantation could lead to better MRI-identified “infill” in patients with an isolated traumatic single full-thickness femoral condylar chondral lesion >2 cm^2^; nevertheless, both groups had very good-to-excellent and good-to-very good chondral repair [64]. This finding indicates that locally dense MSCs after implantation can repair knee cartilage. An MSC injection (plus hyaluronic acid injection) following arthroscopic microfracture implantation led to MRI-identified cartilage repair after 12 months, which is comparable to the effect of an MSC implantation beneath a sutured periosteal patch over the defect [65] (Table 4), which suggests that diffused MSCs after injection also have a potential to repair cartilage defect. A recent study found that isolated autologous adipose-derived MSCs can differentiate toward specific cell types and express extracellular matrix components characteristic for osteo- and chondrogenic lineage. Intra-articular injection using these MSCs with a dose of 5–10 × 10^6^ also led to MRI-identified cartilage repair in patients with knee OA during the 18-month follow-up [66].

Overall, locally dense MSCs after transplantation and diffused MSCs after intra-articular injection (thus without a scaffold) may facilitate cartilage repair/regeneration. These effects can be detected via MRI-identified cartilage volume increase [52,65,67] and directly visualized by arthroscopy [68,69,70].

When opposed to high tibial osteotomy (HTO) + PRP, MSC injection plus HTO + PRP resulted in better pain reduction and arthroscopy-viewed regenerated fibrocartilage [68]. Comparing the effects of intra-articular knee injections of adipose-derived MSCs with different doses (1.0 × 10^7^; 5.0 × 10^7^; 10 × 10^7^ cells) showed that regeneration of hyaline-like articular cartilage can be found by arthroscopy in the high-dose group [70]. Notably, MSC implantation and injection can induce arthroscopy-viewed regenerated cartilage (although the implantation group had better clinical improvements and better cartilage repair) [69].

**Table 4 ijms-24-00059-t004:** Injections vs. surgery and injections.

Cell Type	Cell Dosage	Cell Passage	Combined Interventions	Control with Non-MSC Agents	Knee OA Grading	Injection Time	Follow Up	Outcome	Reference
Autologous ADMSCs	1.2–2.3 × 10^6^	0	Arthroscopic debridement + PRP. Rehabilitation programs were available.	Only arthroscopic debridement + PRP	KL 3.3 ± 0.8 (MSCs group) or 2.7 ± 0.7 (control)	1 (PRP multiple times)	12–18 months	Better symptom relief in the MSCs group. Good results obtained only in young patients and those with early cartilage degeneration.	[9]
Autologous BMMSCs	10 × 10^6^	1	Arthroscopic microfracture + hyaluronic acid injection three times. Individualized rehabilitation programs were available.	MSC implantation beneath a sutured periosteal patch over the cartilage defect.	≥1 symptomatic full-thickness chondral lesion	1	24 months	Both groups had improvements.	[65]
Autologous MSCs (from stromal vascular fraction)	4.11 × 10^6^	0	HTO + PRP	HTO + PRP	KL III or lower	1	14–24 months	HTO + MSCs + PRP resulted in good regenerated fibrocartilage (by arthroscopy) and better pain reduction than HTO + PRP only.	[68]
Autologous ADMSCs	3.19–4.65 × 10^6^	Unavailable	PRP	Implantation vs. injection	KL 1–2; an isolated full-thickness articular cartilage lesion 3.2–9.4 cm^2^	1	24–42 months	MSC implantation resulted in better clinical and second-look arthroscopic outcomes than an MSC injection.	[69]
Autologous BMMSCs	14.6 × 10^6^	1	Hyaluronic acid (3 weeks after HTO + microfracture)	Hyaluronic acid alone (3 weeks after HTO)	Medial OA, KL grading unavailable	1	2 years	Better symptom improvement and cartilage repair (by MRI) were observed in the MSCs group.	[71]

ADMSCs, adipose-derived MSCs; BMMSCs, bone marrow-derived MSCs; HTO, High tibial osteotomy; PRP, platelet-rich plasma; KL, the Kellgren and Lawrence OA classification.

## 6. Do More MSCs Lead to Better Effects?

After injection, MSCs and their secretion may repair damaged issues via diverse mechanisms including chondrocyte differentiation, trophic effects, and immunomodulatory functions. A larger amount of injected MSCs may be expected to induce better effects. Table 5 and Table 6 list dose-comparison studies applying MSC injections in knee OA patients.

In all studies using allogeneic MSCs, lower doses of MSCs, regardless of bone marrow-derived MSCs [41,72] or adipose-derived MSCs [67,73], consistently had better improvements in clinical symptoms and/or MRI-identified cartilage repairs than the higher-dose groups (Table 5).

Even for autologous MSCs, higher-level doses may not necessarily result in better therapeutic effects following intra-articular injections. In six studies using autologous MSCs (Table 6), only two studies detected better effects in the higher-dose groups [42,70]. The results of other studies did not support better effects in higher-dose cases. A study found pain reduction and functional improvement in all treated cases but observed statistical significance only in the lower-dose group; only patients in the higher-dose group had worsened pain and decreased functional scores [74]. In a study with follow-up periods of 12 months [75] and then 4 years [76], the high-dose group had better effects at 12 months following injection, but the lower dose induced a higher level of pain reduction at 4 years. Another study found an MRI-identified increase in cartilage volume in the middle-dose group (Figure 3 in [77]) and the highest level of functional improvement and SF-36 scores at 96 weeks in the middle-dose group, though the highest-dose cases had better scores in some indices [77].

Compared with autologous sources, allogeneic MSCs can be derived from healthy donors and then expanded in vitro to reach clinically relevant numbers. However, allogeneic MSCs derived from bone marrow or adipose can be recognized by the host immune system after injection. Allogeneic MSCs have lower immunogenicity than other allogeneic cell types but these MSCs may induce strong immune responses in vivo and therefore lead to severe consequences [78]. For instance, after performing intra-articular injections of bone marrow-derived equine allogeneic or autologous MSCs, no differences in clinical findings were detected after the first injection; by contrast, adverse responses of the injected joint and an elevation of synovial total nucleated cell counts were found in horses receiving allogeneic MSCs following the second injection [79]. Therefore, a lower dose appears to be suitable for intra-articular injection for knee OA due to cellular and humoral immune responses following injection using allogeneic MSCs.

Patient-derived MSCs (i.e., autologous MSCs) should be safer given that unwanted immune responses can be prevented, but several possible disadvantages hinder the use of high doses. First, knee OA patients are commonly old-aged people, and comorbidities other than knee OA and senescence/age of the patients influence the quality of derived MSCs. For instance, a review analyzing 41 studies emphasized that MSC characteristics and regenerative potential are often affected by cardiovascular disease [80]. In addition to the age of the donors, which may be a parameter greatly affecting MSCs functions, in vitro cell aging (cell passage) during in vitro cell expansion of MSCs can modify cell properties of self-renewal, differentiation, and secretion (autocrine and paracrine) [81]. Second, several local elements in the culture microenvironment may influence MSCs differentiation [82,83]. The in vitro culture expansion process of MSCs may also encounter infection risks [9]. As the result, higher-dose autologous MSCs may hold a greater potential of inducing more serious adverse effects than a lower dose.

In summary, even for autologous MSCs, knee joint injection using more cells may not necessarily result in better effects. Similar findings were also detected in a study using allogeneic and autologous bone marrow-derived MSCs to treat heart diseases, which found the greatest therapeutic effects in the lowest-dose group (20 × 10^6^; 100 × 10^6^; 200 × 10^6^). Based on the abovementioned findings, the most suitable dose for MSCs prepared with a given protocol needs to be studied. However, only a few studies introduced how the adopted dose was determined, such as based on results from animal experiments [74] or previous studies [48,49].

**Table 5 ijms-24-00059-t005:** Dose-comparison studies using allogeneic MSCs.

Cell Type	Cell Dosage	Cell Passage	Combined Interventions	Control with Non-MSC Agents	Knee OA Grading	Injection Time	Follow Up	Outcome	Reference
Allogenic BMMSCs	50 × 10^6^/ 150 × 10^6^	Unavailable	Hyaluronic acid, human serum albumin (1.2%), and plasma-lyte a. Avoid strenuous activities or prolonged weight-bearing for 48 h and running and/or repetitive-impact activity for 6 weeks post-injection.	Hyaluronic acid alone	7–10 days after partial medial meniscectomy	1	2 years	Increased meniscal volume by MRI and pain reduction only in the MSC group (and better in the low-dose group).	[41]
Allogenic ADMSCs	3.9 ×10^6^/ 6.7 ×10^6^	Unavailable	None	Placebo	KL 1–3	1	12 months	Lateral tibial cartilage volume increase by MRI only observed in the low-dose group.	[67]
Allogeneic BMMSCs	25 × 10^6^/ 50 × 10^6^/ 75 × 10^6^/ 150 × 10^6^	Unavailable	Hyaluronic acid	Plasma-lyte a	KL II–III	1	12 months	The trend of pain reduction only observed in the 25 × 10^6^ dose group (but statistically insignificant). Predominant adverse events observed in the higher-dose groups. No MRI improvements.	[72]
Allogeneic ADMSCs	10 × 10^6^/ 20 × 10^6^/ 50 × 10^6^	Unavailable	Rest for 24 h following each injection	None	KL II–IV	2 (0 + 3 weeks)	48 weeks	The low-dose group had better pain reduction and function improvements. MRI assessments showed slight improvements in the low-dose group.	[73]

ADMSCs, adipose-derived MSCs; BMMSCs, bone marrow-derived MSCs; KL, the Kellgren and Lawrence OA classification.

**Table 6 ijms-24-00059-t006:** Dose-comparison studies using autologous MSCs.

Cell Type	Cell Dosage	Cell Passage	Combined Interventions	Control with Non-MSC Agents	Knee OA Grading	Injection Time	Follow Up	Outcome	Reference
Autologous BMAC	≤400 × 10^6^/ >400 × 10^6^	Unavailable	PRP + platelet lysate. After injection, using a knee orthosis and following a weight-bearing protocol.	None	KL I–IV (>50% in early stage, that is, KL I)	1	12 months	Pain reduction and better function observed in both groups. Greater pain reduction occurred in the high-dose group.	[42]
Autologous ADMSC	1.0 × 10^7^/ 5.0 × 10^7^/ 10 × 10^7^	Unavailable	None	None	KL II–IV	1	6 months	Better knee function and pain reduction and reduced cartilage defects by regeneration of hyaline-like cartilage (observed by arthroscopy and MRI) only in the highest dose group.	[70]
Autologous ADMSC	2 × 10^6^/ 10 × 10^6^/ 50 × 10^6^	1	None	None	KL III–IV	1	6 months	Pain reduction and function improvement observed in all cases but statistical significance only observed for the low-dose group.	[74]
Autologous BMMSCs	10 × 10^6^/ 100 × 10^6^	Unavailable	Hyaluronic acid	HA alone	KL II–IV	1	12 months/4 years	12 months: better X-ray and MRI findings only in HA + high-dose group; no effects in the control group./ 4 years: better clinical improvement in high- and low-dose groups. The low-dose group induced higher level of pain reduction.	[75] / [76]
Autologous ADMSC	10 × 10^6^/ 20 × 10^6^/ 50 × 10^6^	4	None	None	KL II–IV	3 (0–6–48 weeks)	96 weeks (≈22.4 months)	Increased cartilage volume by MRI and significant difference detected in the middle-dose group. The middle-dose group also had the highest functional improvement and SF-36 scores at 96 weeks.	[77]
Autologous SVF cells (adipose)	30 × 10^6^ 15 × 10^6^	0	Minimal weight-bearing for 2 days. Full range of motion (non-weight-bearing) was encouraged. Only light activity and previously painful activities should be avoided for the first 3 weeks after injection.	Placebo (zero SVF cells)	KL II–III	1	12 months	Better WOMAC score changes in the high- and low-dose MSCs groups than those in the control (89.5%; 68.2%; 0%). However, no changes in cartilage thickness were detected by MRI.	[84]

ADMSCs, adipose-derived MSCs; BMAC, bone marrow aspirate concentrate, containing MSCs and hematopoietic stem cells (HSCs); BMMSCs, bone marrow-derived MSCs; KL, the Kellgren and Lawrence OA classification; PRP, platelet-rich plasma; SVF, stromal vascular fraction cells; WOMAC, Western Ontario and McMaster Universities Osteoarthritis Index.

Directly comparing MSC doses across studies is unreliable due to age and constitutional differences of donors and inconsistent preparation/culture expansion protocols. Notably, the origin of MSCs probably contributes to the properties of MSCs, such as downstream applications [85]. Thus, a standardized protocol for preparation, culture expansion and evaluation of the biological potentials of MSCs will be helpful and contribute to clinical utility. Essential MSC parameters including surface markers and differentiation potential should be included in the quality control process; for example, surfactome, cartilage-forming capacities, MSC size and granularity, telomere length, senescence status, trophic factor secretion, and immunomodulation feature, can be assessed to evaluate MSC potentials [86].

## 7. Do We Need a Post-Injection Protocol?

Post-injection managements vary across studies. In the study applying MSC injection together with arthroscopic microfracture, an individualized postoperative rehabilitation protocol was provided to each patient. The protocol was designed according to the location and size of where the lesion occurred; for example, patients with patella and trochlea lesions should limit knee flexion during the first several weeks, whereas those with condyle lesions should avoid weight-bearing during the first six weeks [65]. Another study introduced a rehabilitation program composed of supervised progressive resistance training with specific water- and land-based exercises in a 4-month period [87]. Avoiding weight-bearing by using crutches and a range of motion and quadriceps exercises [61], or maintaining minimal weight-bearing by encouraging a full range of motion (non-weight-bearing) and avoiding previously painful activities [84] were also instructed. Conversely, some listed studies did not introduce a post-injection rehabilitation program.

Tailored or individualized post-injection rehabilitation protocol can facilitate recovery of symptoms and cartilage repair due to several considerations.

First, the very low friction between hyaline cartilage surfaces is a fundamental feature of the normal function of knee joints, which originates from the intact lamina splendens, lubricin coating the cartilage surface, and normal elasticity of several cartilage layers relying on the balanced forces from hydrated proteoglycans and collagen fibrils [28]. A trauma or cumulative micro-traumas in knee joints can lead to damage to one or several layers of hyaline cartilage. The damaged integrity of the surface layer increases friction between the hyaline cartilage. Injuries of deeper layers lead to cartilage deformation during loading, which also increases friction between cartilage surfaces. Notably, a 1% increase in the rate of tibial hyaline cartilage loss between baseline and 2 years is associated with a 20% increase in the risk of undergoing knee replacement surgery after 4 years [88]. Injected MSCs may repair the surface to various extents (especially when hyaline cartilage is regenerated). Avoiding high-level weight-bearing during knee movements can prevent harmful impacts on the surface tissue in the process of repair. However, during the non-weight-bearing range of motion, the main stress applied on the cartilage surface is generated in the direction of friction along the surface, that is, the orientation of surface collagen fibrils in normal status; this phenomenon will facilitate forming a normal arrangement of newly regenerated collagen fibrils on the cartilage surface. Sliding between cartilage surfaces can also improve fluid transport into buried cartilage tissues and thus facilitate nutrition supply [89]. As a result, post-injection knee range of motion without weight-bearing needs to be considered.

Second, withstanding loading of knee joints relies on normal morphology and viscoelasticity along the entire thickness of hyaline cartilage, which is damaged in patients with knee OA. Repairing such damages cannot be achieved immediately after MSC injection. Especially, local repair processing in OA patients with focal injuries requires a long duration. Any painful activities of knee joints such as weight-bearing actions may adversely impact the repairing process. Notably, common weight-bearing activities such as walking result in a high level of stress, which is 1–6 MPa, probably up to 12 MPa [90]. However, the osmotic swelling pressure is estimated only at a magnitude of approximately 0.2 MPa (in human femoral heads, and knee cartilage may be at the same magnitude) [31]. Collagen concentration remains steady during aging, but proteoglycan concentration is increasingly dropped [36], which decreases the viscoelasticity of knee cartilage to resist loading in that less amount of water attracted by proteoglycans in the cartilage will diminish the role of water in plasticizing the tissue [91,92].

Third, solute uptake of hyaline cartilage during joint loading is hampered given that the cartilage surface is shielded from synovial fluid [93], which in turn affects the nutrition supply of the cartilage in long-standing, heavy-loaded conditions. Meanwhile, long-duration non-weight-bearing results in cartilage degeneration and should thus also be avoided [31]. Passive diffusion, mechanical “pumping” during dynamic joint loading, and sliding movements during motion induce changes in interstitial fluid pressure to drive exchanges of nutrients and oxygen [89]. Especially, dynamic loading on the cartilage drives solute transport and matrix protein synthesis [94] and may thus modulate the function of chondrocytes. The chondrocyte is one of the potential initiators of knee OA development. Its anabolic and catabolic activities can be inappropriately activated. Abnormal proliferation and apoptosis result in changes in cell numbers. The chondrocytes are exposed to various stimuli including non-physiologic loading, byproducts generated from the destructed matrix, and abnormal levels of cytokines and growth factors [17]. All these functions cannot be altered rapidly following MSC injection, and chondrocytes need a better environment to exert their functions, such as self-renew differentiation, and autocrine and paracrine responses.

Fourth, lower leg muscles play an important role in protecting knee joints. For instance, hip abductor weakness is found to be associated with poor function in knee OA patients [95,96,97]. Impaired hip abductor forces lead to abnormal movement mechanics and joint loading during weight-bearing activities. Conversely, hip abductor strength training may play a beneficial role in protecting knee function [98]. Limb muscles and tendons also can provide a “shock-absorber” mechanism that relieves mechanical energy loaded in the knee joint [99], which indicates that appropriate strength training may be helpful.

Finally, injected MSCs enter a usually hypoxic, inflammatory mediator-rich, and probably low-pH environment in knee OA patients, which is not optimal for not only the survival of MSCs but also in exerting their functions [100]. Structural changes in the cartilage matrix influence chondrocytes. Constituents of the cartilage matrix also interact with cell surface receptors to send signals regulating chondrocyte functions. Several signaling pathways activate various transcription factors, which translocate into the chondrocyte nucleus and regulate the expression of many inflammatory mediators and matrix-degrading enzymes [5]. Thus, physical activities and other factors influencing the local environment should be carefully considered and selected due to such cartilage matrix–chondrocyte interaction and its susceptibility to various causes. For instance, a caloric restriction not only can lower body weight to reduce joint loading but also may induce appropriate autophagy to restore the regenerative capacity of stem cells [101]. Therapeutic modalities, except cold or heat, have seldom been introduced in post-injection programs whereas ultrasound [102], shortwave diathermy [103], transcutaneous electrical nerve stimulation, and interferential currents [104] are beneficial for patients with OA of the knee. Their roles in modulating MSCs are unclear but their effects on pain reduction and symptom improvement indicate their potential to alter the local pathological status of knee joints.

Overall, non- or minimal weight-bearing of the affected knee joint during the early stage after MSC injection can be considered. A tailored protocol consisting of weight-bearing strategies, range of motion, strength exercises, and other therapies can also be instructed to patients.

## 8. Conclusions

Intra-articular MSC injection may improve function and relieve pain for patients with knee OA. Treatments with MSCs derived from different sources have a positive impact on tissue regeneration. Further clinical and in vitro studies are needed to better clarify both the molecular and biochemical mechanisms that MSCs can act alone or in association with PRP or surgical treatments.

The pathological status of knee OA influences the effects of MSC injections. Focal cartilage lesions in OA knees are more likely to respond to an MSC implantation rather than an MSC injection [69], though a large lesion size (>5.7 cm^2^) in the cartilage is still a challenge for implantation therapy of MSCs [105]. Young patients and those with knee OA in early cartilage degeneration are more likely to receive good results following intra-articular injection of MSCs [9].

For MSCs from the same origin, intra-articular allogeneic MSC injection with a lower dose always had better improvements than that with a higher dose, and autologous MSC injection with higher-level doses may not necessarily result in better therapeutic effects than those with lower doses. These results suggest that appropriate MSC doses applied in intra-articular injection to knee OA patients need to be determined for each origin of MSCs.

Compared with fibrocartilage, healthy hyaline cartilage has ultra-low friction during joint motion, which plays a fundamental role in achieving normal knee functions. After MSC injection, regenerated hyaline-like cartilage or fibrocartilage can be observed by arthroscopy, whereas the conditions required for hyaline cartilage regeneration are not understood. Knee OA patients in an early stage and those without large focal lesions are more likely to respond well to intra-articular MSC injection compared with more serious patients. Furthermore, MSC injection combined with other agents such as hyaluronic acid [75] or PRP [58] has better therapeutic effects than MSC injection alone, which implies the possible values of drug cocktail therapy for MSC injection in knee OA patients; types of applied agents and times of injection can be further studied. An individualized post-injection rehabilitation program including joint loading protocol, muscle/tendon functions, and therapeutic modalities may also alter the pathological status of affected knee joints to provide a better environment for local tissue–MSCs and cartilage–matrix interactions. In particular, the magnitude of weight-bearing may play an important role in recovering the anatomical and functional properties of the surface layer of hyaline cartilage.

## Figures and Tables

**Figure 1 ijms-24-00059-f001:**
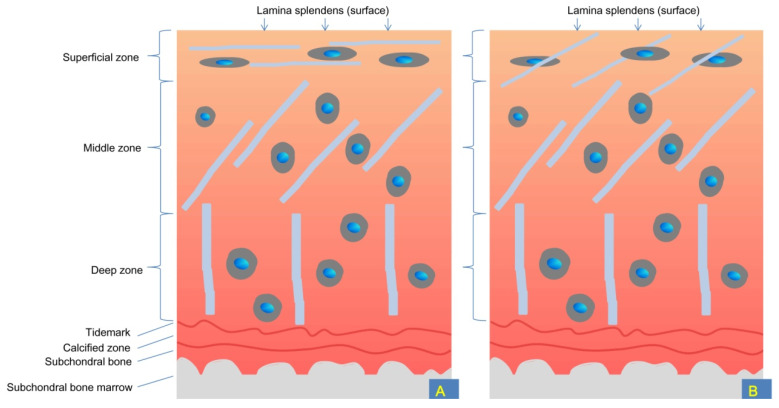
Schematic demonstration of different layers of articular (hyaline) cartilage and subchondral bone of human knee joint. Arrows indicate the outermost layer of articular (hyaline) cartilage, that is, the lamina splendens. In some literature, the lamina splendens is also referred to as “the surface zone.” The superficial zone contains thin collagen fibrils and the middle zone contains thin and thick collagen fibrils; the deep zone contains thick collagen fibrils and is, therefore, the most stress-resistant. (**A**). Traditional concepts suggest that the superficial/gliding zone contains collagen fibrils ***parallel*** to the surface of healthy hyaline cartilage (whereas the lamina splendens does not contain collagen fibrils and chondrocytes); (**B**). Current findings indicate that the superficial/gliding zone contains collagen fibrils ***oblique*** to the surface of healthy hyaline cartilage (and the lamina splendens contain interwoven collagen bundles, which are not shown in this figure). Notably, interwoven collagen bundles within the lamina splendens rarely integrate obliquely oriented collagen fibrils in the superficial zone, which implies no tight connection between the two layers. Such a feature provides limited resistance to tearing/peeling off this surface layer from underlying cartilage tissues during exercise/sports accidents.

## Data Availability

Not applicable.

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
