# Peer review of "Intra-Articular Mesenchymal Stem Cell Injection for Knee Osteoarthritis: Mechanisms and Clinical Evidence"

_ijms, 2022, doi:10.3390/ijms24010059_

Round 1

Reviewer 1 Report

I congratulate the authors on their review paper.  It focuses on areas of importance in regards to MSC therapy in knee OA

 - cell dose

 - cell source

 - pain/function

 - disease modification

The article needs thorough grammatical review.

A combined comparison of injectable therapies and injectable therapies with surgical intervention is not easy to follow.  I would suggest separating these comparisons

 - A) Injectable Therapies comparison

 - B) Surgery (i.e. HTO) + injectable therapies comparison

Section 4 needs revision as it does not have a clear relevance to the review on MSC therapy.  

 - I would consider focussing more on differences that may impact outcome from MSC therapy - i.e. alignment, Grade of OA, weight bearing post MSC therapy - and correlate this with trial protocols.

Author Response

Many thanks for the comments and suggestions.

A grammatical review by native speakers has been performed.

We split the original two tables to six tables. We hope that short tables can be followed more easily. Injections and injection+surgery have been separated.

The aim of Section 4 is to provide a background for knee structure and function relevant to MSC injection and the post-injection rehabilitation protocol in the section “Do we need a post-injection protocol?” We have moved this section to Section 1 and added a paragraph at the end of the Introduction to introduce such aim: “This review starts from discussions of anatomical, biochemical, and biomechanical properties of knee cartilage since these properties are related to what programs should be applied after injection of MSCs”.

The differences of alignment, Grade of OA, weight bearing post-MSC injection between listed studies should have an impact on the outcome of the MSC therapy. However, direct comparisons are difficult due to heterogeneity of these studies and the lacking of many details. Therefore, we only provides several “tendencies” in this review, e.g., “Knee OA patients in an early stage and those without large focal lesions are more likely to respond well to intra-articular MSC injection compared with more serious patients”, and “the magnitude of weight-bearing may play an important role in recovering anatomical and functional properties of the surface layer of hyaline cartilage”.

Reviewer 2 Report

General comments for the authors

In general, the review is difficult to read and follow. The aim is not completely clear considering the text of the manuscript.

The authors want to discuss about too many complex topics in the same review. The result is that each is confused and superficially treated.

As a consequence, the conclusions that the authors claimed are not supported enough.

Therefore, my suggestion is to better focus the review on a specific topic. For example, what is the rationale of the section number 4 (about biomechanics)? It could be a separate review.

Specific comments:

Introduction

Introduction has to be improved better explaining that osteoarthritis is a disease of the whole joint also including infrapatellar fat pad.

Check the whole manuscript adding references where necessary. There are several sentences without any citations.

Please update to more recent references.

A connection between OA features and the use of MSCs is lacking. The authors should also describe the different origin of MSCs (bone marrow, adipose, umbilical, placenta, etc.). Moreover, it should explain that adipose stem cells can be collected from different adipose tissues (subcutaneous, abdominal, infrapatellar fat pad), etc. 

Better define the aims of this review.

Describe the Search strategy of literature.

2. More MSCs lead to better effects?

Studies have to be described in the same order of the table. Otherwise, it is very difficult to read the manuscript.

It should be better to describe separately studies about each type of MSCs.

The authors should justify why they selected these specific studies.

Table 1 is not clear. The rationale of table 2 is not clear. Please clarify the difference between the two tables.

In any case, it is suggested to add a table for each type of MSCs in order to facilitate the reading.

Tables have to be improved better visualizing data.

Please use abbreviations for the different types of MSCs.

For example: bone-marrow MSCs = BMMSCs; adipose derived MSCs = ADMSCs

Explain abbreviations in the footnote of the tables and not inside the line.

Use the plural when referring to MSCs.

Please visualize the studies in the table in the same order of the description in the text.

Control with non-MSC agents is not clear in the table. The term agent is not clear; it is usually referred to chemical and physical agents.

At line 77 the authors started to describe hyaluronic acid comparing its use with MSCs. On this topic only four studies were discussed. This topic is complex and debated and it cannot be limited to few studies also dealing with different types of MSCs.

The same criticism is present in the description of the other substances (PRP and corticosteroid).

The authors described studies about the use of adipose stem cells derived from infrapatellar fat pad. However, it has been reported that this type of stem cells, harvested from patients with osteoarthritis, seems to be primed by the inflammatory joint environment. This point should be discussed.

3. Can MSC injection induce cartilage regeneration?

Section 3 is very brief. The topic is complex and cannot be reduced to a simply short paragraph. Moreover, a new type of MSCs is reported (synovia-derived MSCs).

4. Biomechnics and biochemistry of knee hyaline cartilage and meniscus

In this section the authors described only cartilage and meniscus anatomy and structures.

Considering the aim and the previous part of the review, the reader is expected to find studies on the use of MSCs and biomechanics/biochemistry of these tissues and not only tissues description.

Reviewer 3 Report

The paper has a potential interest considering that cell therapies especially stem cell therapies are now coming of age. I would consider to move all the "anatomical" section at the beginning of the paper just to contextualize the content.

Title of chapter 4 is biomechanics and not biomechnics.

Also consider to have smaller tables. Page-long tables are difficult to read.

Author Response

Many thanks for the comments and suggestions.

The anatomical section has been moved to the beginning of the manuscript.

The word “biomechnics” has been corrected to “biomechanics”.

We have separated the original two tables to six shorter tables.

Round 2

Reviewer 2 Report

In general, the review has been little modified in some parts but it is still difficult to read and follow.

The main criticism is that the authors want to discuss about too many complex topics in the same review. The result is that each is confused and superficially treated.

The studies discussed seem not so comparable as they differ not only for the type of SCs injected but also for combined interventions, control substances, follow-up, rehabilitation protocol, etc.

As a consequence, the conclusions that the authors claimed are not supported enough.

Therefore, my suggestion is to better focus the review on a specific topic.

Author Response

Thank you for the consideration.

The manuscript mainly focuses on effects of doses of MSCs for intra-articular MSC injection for knee OA. Meanwhile, after the literature search, we also acquired some studies that performed other types of comparisons, that is, comparing different substances or different techniques (Injections vs. surgery + injections). These comparisons also provide useful information, for example, “locally dense MSCs after transplantation and diffused MSCs after intra-articular injection (thus without a scaffold) may facilitate cartilage repair/regeneration. These effects can be detected via MRI-identified cartilage volume increase [52,65,67] and directly visualized by arthroscopy [68-70]”, and “MSC implantation and injection can induce arthroscopy-viewed regenerated cartilage (although the implantation group had better clinical improvements and better cartilage repair) [69]”. Especially, the regeneration potential induced by MSC injection is important for this therapy. We therefore place these comparisons in the review as a lead-up of the dose comparison.

An in-depth review focused on a topic is a very good choice but needs a large number of studies in a direction. Tables 5 and 6, as the predetermined focus, contain 11 studies. Such a volume is suitable to be one of several sections in a review. Once more studies are available, a review can specifically center on a topic and form stronger conclusions. For this review, we organize several related issues with current available studies to present several trends/suggestions to the audience for reference.